# A Bacterium-like Particle Vaccine Displaying Envelope Proteins of Canine Distemper Virus Can Induce Immune Responses in Mice and Dogs

**DOI:** 10.3390/v16040549

**Published:** 2024-03-31

**Authors:** Lina Liu, Jianzhong Wang, Ranran Li, Jianzhao Wu, Yongkun Zhao, Feihu Yan, Tiecheng Wang, Yuwei Gao, Zongzheng Zhao, Na Feng, Xianzhu Xia

**Affiliations:** 1College of Veterinary Medicine, Jilin University, Changchun 130000, China; luohuawushengna@163.com; 2Changchun Veterinary Research Institute, Chinese Academy of Agricultural Sciences, Changchun 130122, China; 18653494095@163.com (R.L.); 15054605850@163.com (J.W.); zhaoyongkun1976@126.com (Y.Z.); yanfh1990@163.com (F.Y.); wgcha@163.com (T.W.); gaoyuwei@gmail.com (Y.G.); zzzfusheng@163.com (Z.Z.); 3College of Veterinary Medicine, Jilin Agricultural University, Changchun 130122, China; wjzd2005@163.com

**Keywords:** CDV, F protein, H protein, subunit vaccine, bacterium-like particles, pandas

## Abstract

Canine distemper virus (CDV) can cause fatal infections in giant pandas. Vaccination is crucial to prevent CDV infection in giant pandas. In this study, two bacterium-like particle vaccines F_3_-GEM and H_4_-GEM displaying the trimeric F protein or tetrameric H protein of CDV were constructed based on the Gram-positive enhanced-matrix protein anchor (GEM-PA) surface display system. Electron microscopy and Western blot results revealed that the F or H protein was successfully anchored on the surface of GEM particles. Furthermore, one more bacterium-like particle vaccine F_3_ and H_4_-GEM was also designed, a mixture consisting of F_3_-GEM and H_4_-GEM at a ratio of 1:1. To evaluate the effect of the three vaccines, mice were immunized with F_3_-GEM, H_4_-GEM or F_3_ and H_4_-GEM. It was found that the level of IgG-specific antibodies and neutralizing antibodies in the F_3_ and H_4_-GEM group was higher than the other two groups. Additionally, F_3_ and H_4_-GEM also increased the secretion of Th1-related and Th2-related cytokines. Moreover, F_3_ and H_4_-GEM induce IgG and neutralizing antibodies’ response in dogs. Conclusions: In summary, F_3_ and H_4_-GEM can provoke better immune responses to CDV in mice and dogs. The bacterium-like particle vaccine F_3_ and H_4_-GEM might be a potential vaccine candidate for giant pandas against CDV infection.

## 1. Introduction

Canine distemper (CD) is an acute and highly contagious infectious disease caused by the canine distemper virus (CDV). The main clinical symptoms of CD are fever, diarrhea, vomiting, thickening of the foot pads and secretions in the eyes and nose. CDV can infect dogs and cause their death. In recent years, widespread and multi-host infections of CDV have threatened various wild animals. Recently, there have been reports of fatal infections of CDV in wild animals such as giant pandas, monkey, lions, tigers, leopards, two-toed sloths and coatis [1,2,3,4,5,6]. The infection of CDV in rare wild animals has caused a decline in their population sizes, posing a serious threat to the population security of these species [7,8,9,10]. Furthermore, the CDV infection of furry economically significant animals such as foxes and minks has also resulted in substantial economic losses [11]. It is particularly important to take necessary prevention and control measures for the spread of CDV in animals.

CDV is a single-stranded negative-sense RNA virus which belongs to morbillivirus in the paramyxoviridae family. The genome of CDV contains 15,690 nucleotides, encoding six structural proteins, including nucleocapsid proteins (N), phosphoproteins (P), large proteins (L), matrix proteins (M), hemagglutinin proteins (H) and fusion proteins (F). The N, P, and L proteins form the ribonucleoprotein (RNP) complex with the viral RNA. CDV enters host cells through a complex composed of an H protein tetramer and an F protein trimer. The conformation of H protein and F protein changes after the H protein recognizes specific amino acids. This structural rearrangement in the protein complex helps the virus attach to the surface of the host cell membrane and promote the entry of viral RNP into the cytoplasm [12,13]. The H protein contains epitopes that are recognized by cytotoxic T lymphocytes (CTLs), which can trigger CTL-mediated cellular immune responses. CTL responses play a critical role in protecting the host against CDV infection [14]. Therefore, the H protein could be a targeted gene in the development of CDV genetic-engineering vaccines. Dogs immunized with F proteins obtained stronger protection, especially in limiting the spread of virus infection and animal symptoms [15]. Thus, the F protein is also an important protective antigen for the preparation of CDV subunit vaccines. Currently, vaccination is the primary method for preventing CDV infection, and the selection of protective antigen proteins is the foundation for the successful development of vaccines.

Lactic acid bacteria are commonly found in the intestines of humans and animals and they are extensively used in the food industry. In 2005, Bosma et al. developed a Gram-positive enhancer-matrix protein anchor (GEM-PA) surface display system utilizing non-genetically modified lactic acid bacteria [16]. Hot acid acts on lactic acid bacteria, causing them to lose surface lipophosphate walls, proteins and cytoplasmic contents, but lactic acid bacteria still maintain their original shape and intact peptidoglycans. The above bacterial particles are also called Gram-positive enhancer matrices (GEM). The protein anchor (PA) in this system is derived from the cell-wall peptidoglycan-binding domain located at the C-terminus of the AcmA in lactic acid bacteria. The PA consists of three lysine motifs (LysM), each comprising 45 amino acid residues. Exogenous antigen proteins are specifically bound to the surface of the GEM peptidoglycan skeleton via the PA in a non-covalent manner. The GEM-PA system has the following advantages: first, GEM particles are derived from food-grade Lactococcus lactis, which are non-genetically modified organisms and have no biosafety risks; secondly, the surface of GEM particles can anchor multiple proteins at high density, and it is very simple to purify the binding protein by low-speed centrifugation. Finally, the peptidoglycan component of GEM particles is an agonist of TLR2 and has an auto-adjuvant effect. Thus, this system has been applied to the development of vaccines for bacteria, parasites, viruses and other organisms [17,18,19,20,21,22,23,24].

Vaccination is a primary way to prevent CDV for giant pandas. However, attenuated CDV vaccines provide only limited protection and may pose safety risks, and the level of protective antibodies induced by an inactivated CDV vaccine is insufficient and has a short duration. Therefore, there is an urgent need to develop a safe and effective CDV vaccine designed specifically for giant pandas. In this study, three bacteria-like particles, F3-GEM, H4-GEM and F3andH4-GEM, were prepared. Their immunogenicity was evaluated by analyzing antibodies and cytokines after immunizing mice.

## 2. Materials and Methods

### 2.1. Construction of Recombinant Plasmids and Protein Expression

The F (amino acid 248–605) and H (amino acid 59–607) genes of CDV are from giant pandas isolated in our laboratory (KP793921.1). The recombinant plasmids of F or H genes constructed in this study included a signal peptide, a His-tag and a trimerization motif (T4), or a tetramerization motif (GCN4). Three LysMs sequences in the PA were obtained from GenBank (Gene:U17696.1). To enable an efficient expression of the recombinant protein in a Spodoptera frugiperda (Sf9 insect cells (Invitrogen, Carlsbad, CA, USA), the gene was codon optimized and synthesized by Shengong Biotechnology Company (Shanghai, China). BamHI and KpnI restriction sites were inserted into the pFastBac1 plasmid, resulting in the recombinant plasmids pFastBac1-F_3_-PA and pFastBac1-H_4_-PA being constructed, respectively. The constructed plasmids were transformed into *Escherichia coli* DH10Bac-competent cells to generate recombinant bacmids rBacmidF_3_-PA or rBacmid-H_4_-PA. The recombinant bacmids were transfected into Sf9 cells using the cellfectin II reagent (Gibco, Waltham, MA, USA) at a dose of 6 μg per transfection and incubated at 27 °C for 96 h, and we harvested the first-generation recombinant virus (P1). The virus titer was detected using the bacpak baculovirus rapid titer kit (TaKaRa, Dalian, China) after three generations of passage.

### 2.2. Identification of Recombinant Proteins

The expression of recombinant proteins was identified using an indirect immunofluorescence assay (IFA). Sf9 cells were infected with the recombinant baculovirus rBV-F_3_-PA or rBV-H_4_-PA at 0.5MOI. The Sf9 cells were cultured at 27 °C for 48 h, fixed with 80% cold acetone for 20 min, and washed three times with PBST (Beyotime, Beijing, China). The mouse anti-CDV-H or CDV-F polyclonal antibody (laboratory preservation, no non-specific reaction) diluted with 1% bovine serum albumin (BSA)(Sigma, St. Louis, MO, USA) was added. After the antibody incubation with the cells at 37 °C for 1 h, the cells were washed three times. The 200-fold diluted Cy3-labeled goat anti-mouse IgG (Beyotime, Beijing, China) and DAPI staining solution (Beyotime) were added to the cells and incubated in the dark at 37 °C for 1 h. The cells were observed under a ZEISS fluorescence microscope (Carl Zeiss, Oberkochen, Germany) after washing three times.

Western blotting was performed to identify the expression forms of the proteins. The supernatant and infected cells with the recombinant baculovirus were collected separately. The protein samples were analyzed via SDS-PAGE. The protein was transferred to NC membranes (Merck Millipore, Burlington, MA, USA) and blocked with TBST (Beyotime, Beijing, China) containing 5% skim milk powder (Solarbio, Beijing, China). After blocking, mouse anti-CDV-H or CDV-F polyclonal antibody diluted to 1:200 was added and incubated overnight at 4 °C. The 1:20,000 dilution of HRP-labeled goat anti-mouse IgG (Bioworld, Dublin, OH, USA) were added and incubated at 37 °C for 1 h. The protein bands were analyzed with a Tanon 4600 automatic chemiluminescence image-analysis system after washing the NC membrane 3 times with TBST.

### 2.3. Binding of the Fusion Protein to GEM Particles

GEM particles were prepared as previously reported [25]. Briefly, MG1363 *Lactococcus lactis* (stored in our laboratory) was added to M17 broth containing 0.5% glucose at a ratio of 1:100 and cultured for 15 h at 30 °C. GEM particles were obtained by treating the harvested MG1363 with 10% trichloroacetic acid (TCA), boiling for 30 min, and washing with PBS 5 times. One unit (U) was defined as 2.5 × 10^9^ GEM particles.

In order to determine the maximum binding ability of GEM particles to proteins, the 0.5U GEM particles were incubated with 0 mL, 2 mL, 4 mL, 6 mL and 8 mL of the supernatant after sonication at room temperature for 1 h. Then, the samples were collected and we carried out SDS-PAGE and photographed them with gel image system-analysis software (Tanon). Meanwhile, a calibration curve was generated using BSA protein standards on the same PAGE gel. Finally, the photographs were analyzed using Image J software for the amount of protein bound to the GEM.

The bound GEM particles were identified by Western blot and transmission electron microscopy (TEM). The GEM particles, *Lactococcus lactis* control and F_3_-GEM or H_4_-GEM were centrifuged, and the precipitates were collected and fixed with 2.5% glutaraldehyde for 20 h at 4 °C. The samples were also fixed with 1% osmium tetroxide and dehydrated with 50%, 70% and 90% concentration ethanol successively. The samples were put into the embedding agent and cut into 60–70 nm slices with an ultra-thin microtome and stained with 3% uranium acetate and lead citrate. The GEM particles were observed and photographed using a JEM 1200EXII electron microscope for imaging (JEOL, Tokyo, Japan). F_3_-GEM or H_4_-GEM particles were added to 6 × SDS-PAGE sample loading buffer and boiled for 10 min before SDS-PAGE experiments. The samples then transferred to NC membranes and Western blot experiments were performed according to the previous steps. Finally, GEM particles were stained with SuperSignal^TM^ West Femto (Thermo Scientific, Waltham, MA, USA) and photographed using a Tanon 4600 automatic chemiluminescence image-analysis system (Tanon, Shanghai, China).

### 2.4. Immunization of Mice

Female BALB/c mice aged 6–8 weeks were purchased from Jintai Meidi Biotechnology Co., Ltd. (Changchun, China). ADJ-802 adjuvant was obtained from Aidejia Biotech Co., Ltd. (Zhengzhou, China). The BALB/c mice were randomly divided into five groups, each group consisting of ten mice. The hind legs of the mice were injected intramuscularly (IM) with F_3_-GEM or H_4_-GEM mixed with ADJ-802 adjuvant, with a volume of 100 μL per mouse. The mice were immunized with 5 μg, 15 μg or 30 μg CDV-F protein or CDV-H protein, respectively, and the control group was immunized with GEM particles and PBS. All the groups were boosted twice at the 2nd and 4th weeks after the initial immunization. Blood samples were collected at the 2nd, 4th, 6th and 7th weeks after the initial immunization, and the serum was separated and stored at −20 °C after being heat-inactivated at 56 °C for 30 min. The optimal immunization dose was determined through an indirect ELISA to detect serum-specific IgG antibodies.

The BALB/c mice were divided into five groups, each containing ten mice. The mice were immunized with F_3_-GEM, H_4_-GEM, and a mixture of F_3_-GEM and H_4_-GEM at a 1:1 ratio (F_3_ and H_4_-GEM). The immunization dose used was based on the determined optimal dose. The control group was immunized with PBS or GEM particles. Boosted immunizations were performed on the 2nd and 4th weeks after the initial immunization. Blood samples were collected at the 2nd, 4th, 6th and 7th weeks after the initial immunization. The spleens of the mice were taken at the 5th week after immunization. The serum was separated and stored at −20 °C after being heat-inactivated at 56 °C for 30 min. The effectiveness of the vaccine immunization was determined by measuring IgG antibodies and neutralizing antibodies. The levels of specific IgG antibody titers and antibody subtypes in the serum of immunized mice were detected via an indirect ELISA. The proteins of CDV-F and CDV-H stored in our laboratory were diluted to a concentration of 2 μg/mL with the coating buffer. In total. 100 μL of each diluted protein was added to the 96-well polystyrene microtiter plates (Corning Costar) and incubated overnight at 4 °C. Additionally, 3% skimmed milk powder was added and blocked at 37 °C for 2 h after washing the plate with PBST three times. The serum was diluted with 1% skimmed-milk powder, double the ratio diluted from 1:80, 100 μL per well, and incubated at 37 °C for 1 h. After washing with PBST, 100 μL goat anti-mouse IgG HRP (Bioworld) diluted 20,000 times was added to each well and incubated at 37 °C for 1 h. After washing the microplate plates with PBST for the fourth time, TMB was added and placed in the dark for 10 min. The OD_450_nm value was read by an ELISA plate reader (Bio-Rad) after the microplate added the termination solution (beyotime). The positive criterion was that the OD_450_nm value of the tested sample was more than 2 times higher than that of the negative serum. HRP-labeled sheep anti-mouse IgG1 antibodies (Southern Biotech) or HRP-labeled sheep anti-mouse IgG2a antibodies (Southern Biotech) were used as the secondary antibodies. The antibody subtype was analyzed using the ratio of IgG2a to IgG1.

### 2.5. Antibody Neutralization Test

To determine the neutralizing antibody titers in mouse serum, a neutralization assay was conducted. The serum was serially diluted, starting from a four-fold dilution. Then, 100 TCID50 CDV-eGFP, a recombination with enhanced green fluorescent protein (laboratory preservation), was added to each well. The plate was incubated in a 5% CO_2_ incubator at 37 °C for 1 h. Then, 2 × 10^4^ vero cells were added to each well incubated for 4–5 days. Finally, the titer of the neutralizing antibody was calculated by observing the green fluorescence.

### 2.6. Specific Antigen Immune Response

IL-4 and IFN-γ cytokines in the spleen cells of mice were measured through a mouse enzyme-linked Immunospot (ELISpot) assay kit (MABTECH). The spleen cells of mice were collected and prepared as spleen cell suspensions at 5w after immunization. Additionally, 2.5 × 10^6^ cells/mL spleen cells and 10 μg/mL CDV-F or CDV-H protein, which was a stimulant, as added to the pre-coated plates. The wells which were added to ConA were positive controls, and those which were not added to the stimulant were negative controls. The plates were incubated in a 5% CO_2_ incubator at 37 °C for 36 h. 1 μg/mL IL-4, or IFN-γ antibodies were added to the plates and incubated at room temperature in the dark for 2 h. The plates were washed again and incubated with antibodies against Streptavidin-HRP. The spots appeared after adding TMB. The number of spots was counted using the Mabtech IRIS FluoroSpot/ELISpot reader.

### 2.7. Cytokine Detection

The spleens of mice were collected at 7 days after the third immunization and prepared into spleen cell suspensions. Three mice per group were included. In total, 2.5 × 10^6^ spleen cells was added to a 12-well cell-culture plate. In total, 10 μg/mL CDV-F or CDV-H protein, which was a stimulant, was also added to the plates. The wells, to which were added ConA, were positive controls, and those to which the stimulant was not added were negative controls. The plates were incubated in a 5% CO_2_ incubator at 37 °C for 60 h. IFN-γ, IL-2, TNF-α, IL-4, IL-6 and IL-10 in the supernatant were evaluated by V-PLEX Human Cytokine 30-Plex Kit with measurements by Meso QuickPlex SQ120 (Meso Scale Discovery, Rockville, MD, USA).

### 2.8. Immunization of Dogs

Twenty-five 3-month-old beagle dogs (negative for both CDV and anti-CDV antibodies) were divided into five groups with five dogs in each group. The hind legs of the dogs were injected intramuscularly (IM) with F_3_-GEM, H_4_-GEM and F_3_ and H_4_-GEM mixed with ADJ-802 adjuvant, with 400 μg of protein per dog. Both the GEM and PBS groups were used as negative controls. Boosted immunizations were performed on the 2nd and 4th weeks after the initial immunization. Blood samples were collected at the 2nd, 4th, 6th and 8th weeks after the initial immunization. The serum was separated and stored at −20 °C after being heat-inactivated at 56 °C for 30 min. The effectiveness of the vaccine immunization was determined by measuring IgG antibodies and neutralizing antibodies.

### 2.9. Statistical Analysis

Statistical analysis was conducted using GraphPad Prism 8.3.0 software. The results are expressed as mean ± SD. Significance differences between the groups were analyzed using a one-way ANOVA. * denotes statistical significance. * *p* < 0.05, ** *p* < 0.01, *** *p* < 0.001 and **** *p* < 0.0001.

### 2.10. Laboratory Facility and Ethics Statement

All research procedures adhered to the “Ethical Principles for Animal Experimentation” (GB 14925-2001) and were approved by the Animal Ethics Committee of Changchun Veterinary Research Institute, Chinese Academy of Agricultural Sciences (Laboratory Animal Care and Use Committee Authorization, permit number JSY-DW-2023-05).

## 3. Results

### 3.1. Expression of F_3_-PA and H_4_-PA Proteins

The design strategy for the expression of F_3_-PA and H_4_-PA is illustrated in Figure 1A. The trimerization motif (T4) allows the F protein to form trimers, while the tetramerization motif (GCN4) enables the H protein to form tetramers. Both the F and H proteins can specifically bind to the GEM peptideglycan scaffold through the protein anchor (PA) in a non-covalent manner. We expressed the F and H proteins separately by using the Bac-to-Bac™ Baculovirus Expression System. An indirect immunofluorescence assay showed that both F and H proteins were expressed (Figure 1B). Western blot experiments showed that the bands of F_3_-PA (80 kD) and H_4_-PA (110 kD) proteins were detected in the supernatant after sonication (Figure 1C). Meanwhile, the addition of a non-reducing loading buffer showed that the F and H formed trimers and tetramers, respectively (Figure 1D). These results indicate the F and H proteins of CDV were successfully expressed and formed trimers and tetramers, respectively. Sf9 cells were infected with recombinant baculovirus rBV-F_3_-PA and rBV-H_4_-PA, and the supernatant was collected after 4 days of culture at 27 °C. The virus titer of the third generation was identified after two successive generations. The results showed that the titers of rBV-F_3_-PA and rBV-H_4_-PA were 1.10 × 10^8^ and 1.08 × 10^8^, respectively (Figure 1E).

### 3.2. Binding and Identification of the F or H Protein with GEM Particles

To determine whether the *Lactococcus lactises* forms GEM particles after treatment, the treated and untreaded *Lactococcus lactises* were made into frozen sections and we observed their morphology under a transmission electron microscope. The untreated *Lactococcus lactises* exhibited a darker cytoplasm color and contained more contents (Figure 2A). Conversely, the treated Lactococcus lactises showed reduced content while maintaining the bacterial skeleton and morphology (Figure 2B). This result indicated the GEM particles were successfully prepared.

To confirm the binding between F or H proteins with GEM particles, we initially observed GEM particles which might bind to F or H protein under transmission electron microscopy. Filamentous substances were exhibited on the surface of GEM particles (Figure 2C,D). Furthermore, to further validate the binding, the Western blot experiment was performed. We found the GEM particles which bound F or H proteins specifically reacted with mouse anti-CDV-H or CDV-F polyclonal antibody (Figure 2E). The above results indicated that the F or H protein could bind to the surface of GEM particles.

### 3.3. Determination of the Maximum Binding Capacity of the F or H Protein with GEM Particles

To determine the maximum binding capacity of GEM particles to F or H protein, 0.5U of GEM particles incubated with different volumes of F or H proteins was observed. The results demonstrated that the amount of binding protein increased with the increase in volumes of F or H proteins. The binding amount of 1U GEM particles reached saturation when the volumes of F and H proteins were 8 mL and 12 mL, respectively (Figure 2F,G). BSA at various concentrations was used as the standard protein for SDS-PAGE analysis, and a standard curve was generated using Image J software for further analysis. Based on our calculations, the binding capacities of F and H proteins to 1U of GEM particles were 157 μg and 215 μg, respectively (Figure 2H).

### 3.4. Determination of IgG-Specific Antibody in Serum

To assess the immunogenicity of F_3_-GEM and H_4_-GEM, mice were immunized with F_3_-GEM and H_4_-GEM according to the procedure described in Figure 3A. The optimal immunization dosage of F_3_-GEM or H_4_-GEM was determined to detect the levels of specific IgG antibodies in the serum of immunized mice via an indirect ELISA. The results indicated that all immunized mice exhibited IgG-specific antibodies against F or H proteins. The antibody levels increased in correlation with the immunization dosage and displayed a dose-dependent manner. The serum antibody levels of mice immunized with 15 µg of protein were significantly higher than those immunized with 5 ug of protein (*p* < 0.0001 in the F_3_-GEM group and *p* < 0.01 in the H_4_-GEM group), but there was no difference in the levels between mice immunized with 15 μg and 30 μg protein (Figure 3B,C). Consequently, the optimal immune dose of F_3_-GEM and H_4_-GEM was 15 ug.

In order to compare the levels of specific IgG antibodies in the serum of mice induced by F_3_-GEM, H_4_-GEM or F_3_ and H_4_-GEM, the levels of IgG antibodies were detected. The results revealed that the levels of IgG antibodies in all immunized groups were significantly increased compared with the control group, but the level of IgG in F_3_ and H_4_-GEM group was significantly higher than that in F_3_-GEM and H_4_-GEM groups (** *p* < 0.01, *** *p* < 0.001 and **** *p* < 0.0001) (Figure 3D). This suggests that F_3_-GEM, H_4_-GEM, and F_3_ and H_4_-GEM can induce IgG antibodies in mice, but F_3_ and H_4_-GEM is the best.

The ratio of IgG2a to IgG1 is usually used to evaluate the Th1 or Th2 immune response induced by vaccines. The Th1 and Th2 responses produce IgG2a and IgG1 antibodies, respectively. The levels of specific IgG antibody subtypes in the sera of mice were determined via an ELISA. The IgG2a/IgG1 ratios in the F_3_-GEM, H_4_-GEM and F_3_ and H_4_-GEM immunization groups were all less than 1.0 (Figure 3E,F), indicating that all the three vaccines could induce a Th2-biased immune response.

### 3.5. Determination of Neutralizing Antibody Levels

To assess the level of neutralizing antibody after vaccination, the neutralizing antibody titers were determined using a CDV-eGFP virus constructed in our laboratory. The results indicated that all the three vaccines induced neutralizing antibodies compared to the control group (PBS or GEM group). The average titers of neutralizing antibodies in the F_3_ and H_4_-GEM group were higher than those in the F_3_-GEM and H_4_-GEM groups (Figure 3G), suggesting F_3_ and H_4_-GEM has better immunogenicity.

### 3.6. Detection of Cytokines Secreted in Splenocytes

The levels of IFN-γ and IL-4 secreting via spleen lymphocytes were determined using ELISpot. The mice vaccinated with F_3_-GEM, H_4_-GEM or F_3_ and H_4_-GEM were able to generate cells that specifically secreted IFN-γ and IL-4. The IFN-γ and IL-4 in the immune groups were significantly higher than that in the PBS or GEM group (Figure 4).

To further evaluate the immune responses induced by the three vaccines, the concentrations of Th1-related cytokines (IFN-γ, TNF-α and IL-2) and Th2-related cytokines (IL-4, IL-6 and IL-10) were measured. The results demonstrated that the level of IFN-γ, TNF-α, IL-2, IL-4, IL-6 and IL-10 in the immunized groups were significantly higher compared to that in the control groups (PBS and GEM), but cytokines in the F_3_ and H_4_-GEM group exhibited the highest concentration (Figure 5). These results suggested the three groups induced a high level of Th1-related and Th2-related cytokine secretion (* *p* < 0.05, ** *p* < 0.01, *** *p* < 0.001 and **** *p* < 0.0001).

### 3.7. Immune Response in Dogs

To further evaluate the immune effects of the three vaccines, IgG antibodies and neutralizing antibodies were also measured in immunized dogs. The results revealed that the levels of IgG and neutralizing antibodies in all immunized groups were significantly increased compared to the control groups (PBS and GEM), and the F_3_ and H_4_-GEM group exhibited higher levels of IgG and neutralizing antibodies than the other two groups (Figure 6).

## 4. Discussion

CDV can not only infect canines but also wild land animals and aquatic animals, which brings a threat to endangered wildlife populations. In China, CDV is predominantly found in domestic dogs, from which it spreads to infect giant pandas, panthera tigris altaica, ailurus fulgens, monkeys, minks and foxes. It was reported that CDV caused lethal infections in five giant pandas at the Rare Wildlife Rescue and Feeding Research Center in Shanxi Province, China [26], highlighting the urgent need for the prevention and control of CDV in giant pandas. The commercial vaccines available for CDV are usually multiple inactivated vaccines or attenuated vaccines developed for pets, which are unsuitable for the vaccination of giant pandas. The CDV-inactivated vaccines have advantages of high safety and simple preparation, but the antibody titer is not high and their protection period is short. The replication of attenuated vaccines in immunized animals may lead to reverse viral virulence, resulting in death and the spread of the virus by immunized animals [27]. Based on these considerations, we conducted research on CDV vaccines specifically for giant pandas.

The CDV encodes six structural proteins; the fusion protein F and the hemagglutinin protein H are important viral envelope glycoproteins. The F protein possesses group-specific antigenic epitopes that can induce immune responses to prevent virus infection. The H protein serves as the primary immunoprotective antigen, triggering the production of neutralizing antibodies in the host. Both F and H proteins are ideal targets for vaccine development. The F and H proteins exist in trimer and tetramer forms in their natural states, respectively. When the virus encounters the host cell, the F and H proteins form a protein complex, and the protein complex undergoes conformational rearrangement during membrane fusion [28,29]. Natural proteins generally exhibit better immunogenicity than monomeric proteins. In this study, we introduced the motif T4 or GCN4 into the recombinant plasmid to promote the formation of the trimeric F protein or tetrameric H protein [30,31,32]. The F or H protein is connected the PA gene through a flexible linker (GGGGS). The linker plays a crucial role in ensuring the stability and biological activity of the F or H protein [33]. The flexible linker guarantees that PA can bind at the appropriate position of GEM and maintain the stability of the structure of the F or H protein without compromising its immunogenicity. Our results demonstrate that the F protein and H protein form trimers and tetramers, respectively, and successfully anchored on GEM particles.

It was reported that ferrets immunized with F protein and H proteins had better immune effects than ferrets immunized with the F protein or H protein alone [34]. A recombinant canarypox virus ALVAC-CDV-H-F vaccine expressed that H and F proteins could also protect against CDV infection [35]. Thus, in this study, we designed one more immunization group, F_3_ and H_4_-GEM, which consisted of a mixture of F_3_-GEM and H_4_-GEM at a protein content ratio of 1:1. The results demonstrated that the levels of IgG and neutralizing antibodies in the serum of mice and dogs in the F_3_ and H_4_-GEM group were higher than those in the F_3_-GEM or H_4_-GEM groups. We speculate that this might be related to the fact that both the F and H proteins had immunogenicity. Our results also confirmed that the F and H proteins play an important role in resistance to CDV.

The GEM-PA system allows for the anchoring of exogenously added antigens onto GEM particles. GEM particles could interact with antigen-presenting cells on the mucosal surface and promote antigen capture, which could induce antigen-specific responses of Th cells, cytotoxic T lymphocytes and IgA-secreting B cells and initiate both local mucosal and systemic immune responses [36,37]. GEM particles can also be combined with different antigen proteins to provide a good strategy for the development of multivalent vaccines [16]. Moreover, GEM particles can induce the maturation of host dendritic cells (dendritic cellDCs) and secrete cytokines by activating the TLR2 signaling pathway, thereby improving humoral and cellular immune responses [38]. It was reported that a high level of neutralizing antibodies could be detected after the immunization of African wild dogs with an inactivated vaccine against CDV, but it was not clear whether it could induce cellular immunity [39]. In this study, all dogs immunized with GEM particle vaccines could induce the secretion of Th1-related and Th2-related cytokines. Our results provide further evidence for the advantages of using the GEM-PA display system to develop vaccine-induced immune responses.

It was reported that the neutralizing antibody titers of CDV-infected dogs ranged from 64 to 256 [40]. In our study, the neutralizing average antibody titer of immunized dogs was about 8, which is much lower than that of CDV-infected dogs. A study reported that the neutralizing antibody titers of CDV DNA vaccine-immunized dogs ranged from 8 to 32, and 60% (3/5) of immunized dogs had a neutralizing antibody titer of 8. Although the antibody titer of immunized dogs was much lower than that of infected dogs, all immunized dogs were protected after the CDV challenge [40]. This may be related to the important role of cellular immunity in CDV immunization.

## 5. Conclusions

In conclusion, our study successfully developed a bacterium-like particle vaccine displaying the trimeric F protein or tetrameric H protein of CDV. The results clearly demonstrated that the bacterial-like particle vaccine F_3_ and H_4_-GEM can effectively induce immune responses in mice and dogs. The product F_3_ and H_4_-GEM displays an excellent immunogenicity and might be a potential vaccine candidate for giant pandas against CDV infection. The protective efficacy of F_3_ and H_4_-GEM in animal models will be evaluated in future studies.

## Figures and Tables

**Figure 1 viruses-16-00549-f001:**
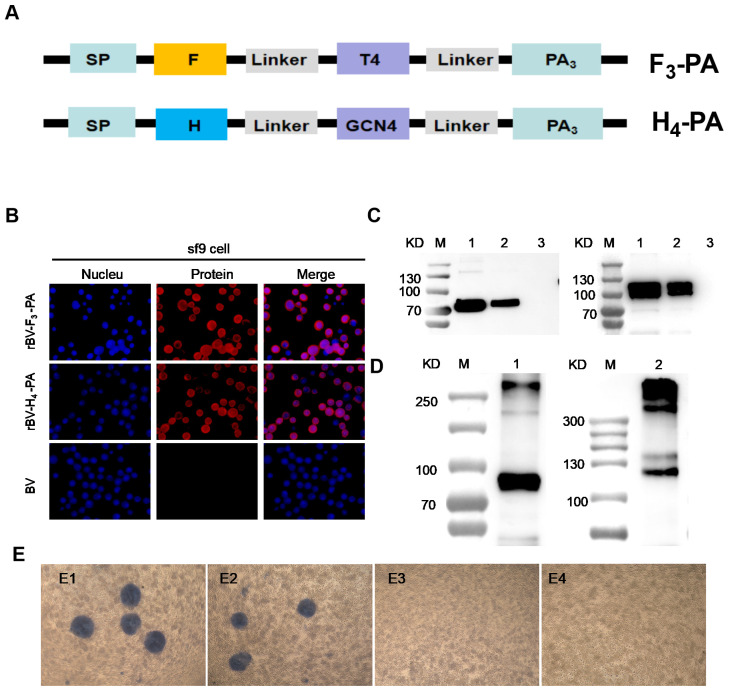
Construction and identification of F_3_-PA and H_4_-PA. (**A**) Schematic of the recombinant baculovirus expressing the F_3_-PA and H_4_-PA proteins. SP: signal peptide, T4: the trimeric motif; GCN4: the tetrameric motif; linker: (GGGGS)_3_; PA: indicates protein anchor. (**B**) IFA detection of F_3_-PA and H_4_-PA expression in baculovirus-infected Sf9 cells. Cells were infected with rBV-F_3_-PA or rBV-H_4_-PA. After 48 h, the cells were detected with a mouse anti-F or H protein polyclonal antibody. (**C**) Western blot analysis of F or H protein expression in Sf9-infected cells. M: molecular-weight marker, 1: cell sedimentation, 2: supernatant after sonication, and 3: uninfected cells. (**D**) Western blot identification of F_3_ and H_4_ proteins with non-reduction treatment; M: molecular-weight marker, 1: supernatant after sonication of F_3_-PA; 2: supernatant after sonication of H_4_-PA; (**E**) and analysis of the recombinant baculovirus titers. (**E1**): F_3_-PA; (**E2**): H_4_-PA; (**E3**,**E4**): Baculovirus control.

**Figure 2 viruses-16-00549-f002:**
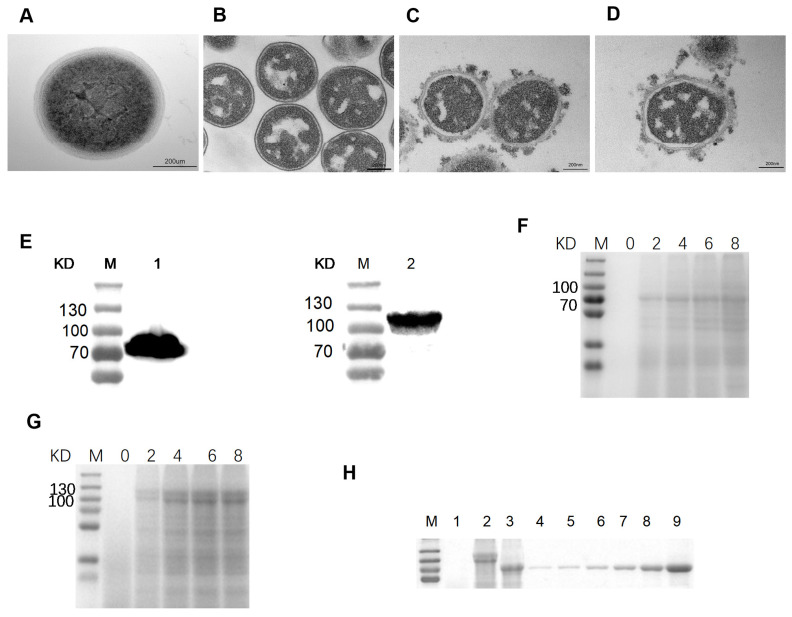
Identification of F3-GEM and H4-GEM. (**A**) TEM images of MG1363 *Lactococcus lactis*. (**B**) TEM images of GEM particles. (**C**) TEM images of F_3_-GEM. (**D**) TEM images of H_4_-GEM. (**E**) Western blot identification of F or H protein anchored on GEM particles. 1: F_3_-GEM; 2: H_4_-GEM. (**F**,**G**): 0.5 U GEM particles were mixed with 0 mL, 2 mL, 4 mL, 6 mL and 8 mL F or H protein. (**H**) The amount of F or H protein was determined. 1: GEM; 2: H_4_-GEM; 3: F_3_-GEMl; and 4–9: different concentrations of BSA proteins.

**Figure 3 viruses-16-00549-f003:**
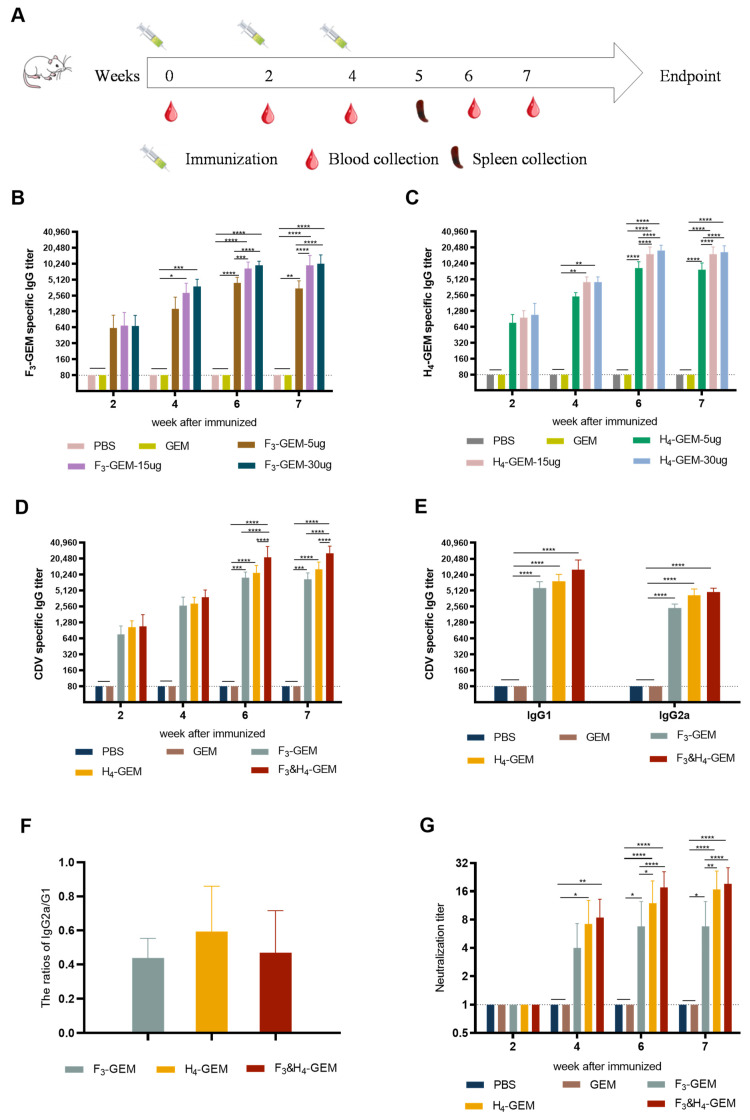
Serum antibody responses induced by F_3_-GEM and H_4_-GEM. Serum samples were collected at weeks 0, 2, 4, 6 and 7. Mouse specific total IgG, IgG1 and IgG2a antibody responses were measured using end-point dilution titers (*n* = 10 mice/group/time point). Data are shown as the mean ± SD and were analyzed by one-way ANOVA (* *p* < 0.05, ** *p* < 0.01, *** *p* < 0.001 and **** *p* < 0.0001). (**A**) Schematic of the experiment. (**B**,**C**) Analysis of serum specific total IgG titers induced by different doses of F_3_-GEM or H_4_-GEM. (**D**) The IgG titers induced via F_3_-GEM, H_4_-GEM or F_3_ and H_4_-GEM. (**E**): Analysis of IgG1 and IgG2a titers induced by the three vaccines. (**F**) The ratios of IgG2a/IgG1. (**G**) The neutralization antibody titers of the three vaccines.

**Figure 4 viruses-16-00549-f004:**
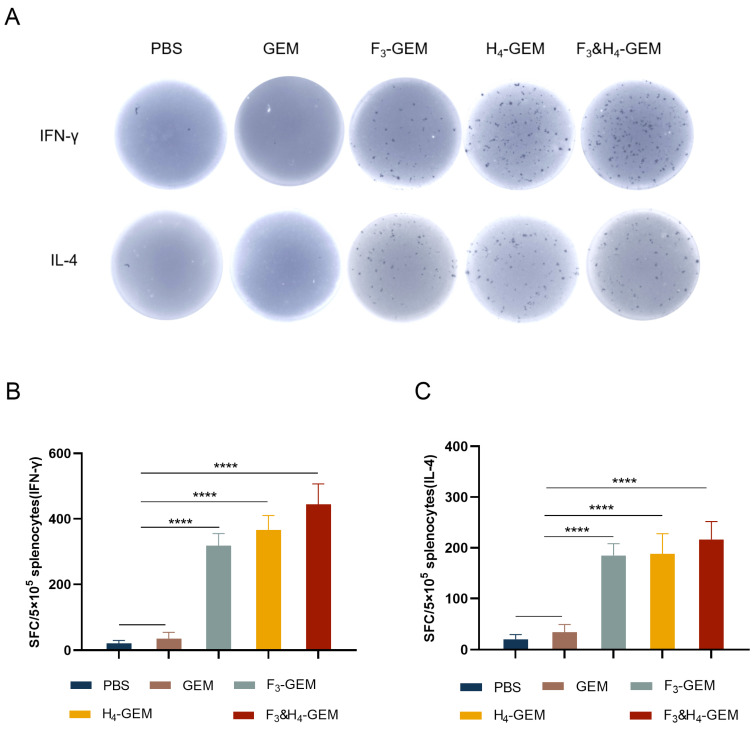
ELISpot analysis of IFN-γ and IL-4 secretion via mouse splenocytes. The splenocytes were collected from each group 7 days after the third immunization was treated and analyzed. (**A**) The spots on the pictures showed the positive cells that secrete cytokines. The secretions of IFN-γ (**B**) and IL-4 (**C**) were measured by using an ELISpot kit. Data are shown as the mean ± SD and were analyzed using one-way ANOVA (**** *p* < 0.0001).

**Figure 5 viruses-16-00549-f005:**
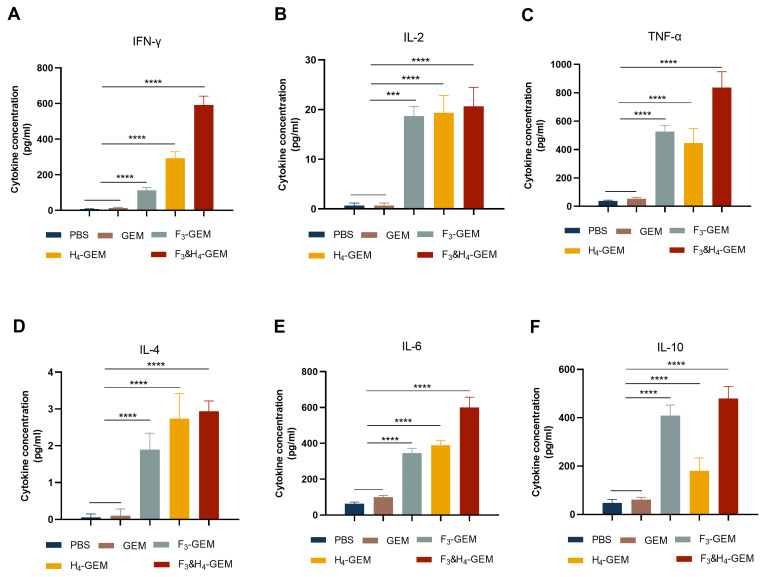
Detection of cytokines in splenocytes. Splenocytes were harvested from the mice 7 days after the third immunization and restimulated with F or H (10 µg/mL) in culture ex vivo. The concentrations of IFN-γ (**A**), IL-2 (**B**), TNF-α (**C**), IL-4 (**D**), IL-6 (**E**) and IL-10 (**F**) in the supernatant were measured with Meso Scale Discovery based on electrochemiluminescence. Data are shown as the mean ± SD and were analyzed using one-way ANOVA (*** *p* < 0.001 and **** *p* < 0.0001).

**Figure 6 viruses-16-00549-f006:**
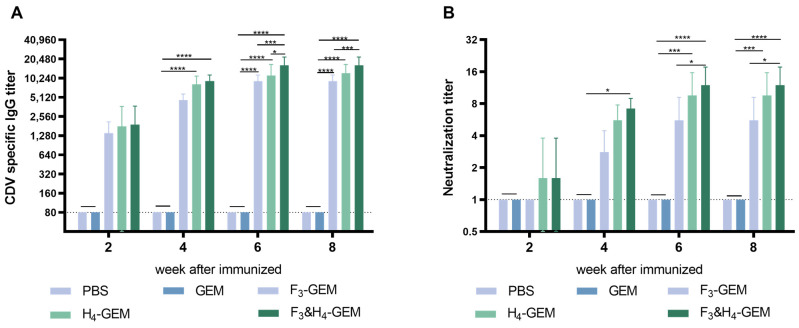
Antibodies in the immunized dogs. Data are shown as the mean ± SD and were analyzed by one-way ANOVA (* *p* < 0.05, *** *p* < 0.001, **** *p* < 0.0001). (**A**) The IgG-specific antibody titers in the serum of dogs. (**B**) The neutralization antibody titers in the serum of dogs.

## Data Availability

The data underlying this article cannot be shared publicly for the privacy reason in the study. The data will be shared on reasonable request to the corresponding author.

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
