# Peer review of "A Bacterium-like Particle Vaccine Displaying Envelope Proteins of Canine Distemper Virus Can Induce Immune Responses in Mice and Dogs"

_viruses, 2024, doi:10.3390/v16040549_

Round 1

Reviewer 1 Report

Comments and Suggestions for Authors

In this manuscript “A bacterium-like particle vaccine displaying envelope proteins of canine distemper virus can induce immune responses in mice and dogs” by Liu et al., the author have used GEM based modality to develop Canine distemper virus (CDV) vaccine. CDV is highly contagious and can cause potentially fatal disease in dogs and other carnivores.

Specific Comments:

1.       Abstract: Subtitles in abstract can be removed, and abstract structuring can be improved.

2.       Introduction: This vaccine modality is interesting, however describing advantages of GEM modality over pre-existing vaccine would enable the readers to better appreciate the author’s main goal.

3.       Experimental design:

a.       Since CDV transmission is through aerosol, why GEM-vaccine is given IM? It would be better to test IN route.

b.      What is the level of free or unbound F3, and H4 recombinant protein in F3-GEM and H4-GEM preps? What are the measures to remove the unbound F3 or H4 proteins during the GEM preps? Including a control group, immunization with only recombinant proteins, would enable it to differentiate the advantage of GEM-prep based vaccine.

4.       Methods: Subsection 2.3 and 2.4 can be combined.

5.       Results:

a.       What is the IgA levels?

b.      Why Nt titer is <32 in immunized mice? Does CDV induce this low Nt titers? It would be beneficial if authors could discuss more on Nt Ab levels during CDV infection and/or trend seen during post-vaccination with existing vaccine. This would enable the reader to understand Nt Ab mediated correlate of protection.

c.       What is the difference in Ab levels, post-vaccination in GEM-based vaccine preps vs existing CDV-vaccine, in dogs? Discussing about this point would be helpful to show advantage of GEM-based vaccine.

d.      Fig.2H: Is there any other method to quantify bound vs free F3 and H4 in preps? As current method uses denaturing SDS-PAGE, it will detect all protein content in the prep, hence it is unclear how it distinguishes bound vs free F3 / H4 in preps.

e.      Fig 6B: Please check the y-axis title.

Comments on the Quality of English Language

Spelling check and minor grammar corrections are needed.

Reviewer 2 Report

Comments and Suggestions for Authors

I have read through this quite comprehensive paper and am impressed by the technology,  by the diligent testing of the product as it was being made,  and then its testing in mice and dogs.   The overall results show that the authours have produced an antigenic product that induces several in-vitro measures of specific immunity. 

The idea of producing hypersafe vaccines that induces effective protective immunity in rare wild animals is not new but remains worth exploring and I  appreciate what the authors are trying to achieve. However, as will be clear from my comments below I have reservations about its practicality for use in wild animals.

As a virologist and immunologist, I can follow most of these aspects of the submission and have little comment. I have no experience of the “bacterium-like” particle and cannot really pass judgement on this aspect of the work. Therefore, what follows are more general questions.

1). Can the authours explain why they did not challenge the immunized dogs with virulent CDV? This surely needed to be done and seems to me a serious omission.

2).  The emphasis in this paper seems to be about producing a vaccine to use in giant pandas. Surely, however, such a vaccine might be of great value in other rare and endangered susceptible species such as the Siberian Tiger. Could the authours consider this broader approach or explain why they are so panda focussed.

3) In many other parts of the world wildlife suffer from frequent introductions of infection with CDV from neighbouring domestic dogs. This is combatted in several countries in Africa and Asia by specifically focussed programmes to routinely immunize domestic dogs living close to or intermingling with rarer wild species. I am sure the authours probably know all about this. can they tell us if this basic control method being used in China, and if it isn’t, why can it not be considered together with the proposed vaccination of the vulnerable species themselves? 

4) Can the authours describe why they feel it might be better to take the risk of vaccinating the rare wild animals themselves rather than vaccinating surrounding dogs?

5) In the Discussion (lines 403 and 404) the sentence “In China, CDV is predominantly found in giant pandas, panthera tigris altaica, ailurus fulgens, monkeys, minks and foxes” is incorrect. More likely in China is that CDV is predominantly found in domestic dogs from which it spreads to infect giant pandas, tigers etc etc…”     

6). I assume this bacterium-like particle is inert and therefore this is an inactive vaccine.  If so, then the immunity it produces may very well be short-lived. This would be disadvantageous for use in wild animals which you would not wish to restrain annually for vaccination. Nor would you want a vaccine that needs to be inoculated into a wild animal three times in four weeks for primary immunization, as was reported here . How do you propose to use this dead vaccine in the wild animals that are the targets of this work? How will you actully administer the vaccine into wild giant pandas or a wild tiger, or even into a captive tiger?

I think the paper will have a broader appeal and prove more useful if the authours consider and address points 1) to 6) above.

Comments on the Quality of English Language

Minor editing of English language required.
